# Conditional knockdown of transformer in sheep blow fly suggests a role in repression of dosage compensation and potential for population suppression

Megan E. Williamson⬥, Ying Yan⬥¤, Maxwell J. Scott⬥*

Department of Entomology and Plant Pathology, North Carolina State University, Raleigh, North Carolina, United States of America

¤ Current address: Department of Insect Biotechnology in Plant Protection, Institute for Insect Biotechnology, Justus-Liebig-University Giessen, Giessen, Germany

* mjscott3@ncsu.edu

**Data Availability Statement:** The authors confirm that all data underlying the findings are fully available without restriction. All relevant data are within the paper and its Supporting Information

## Abstract

The *transformer* (*tra*) gene is essential for female development in many insect species, including the Australian sheep blow fly, *Lucilia cuprina*. Sex-specific *tra* RNA splicing is controlled by *Sex lethal* (*Sxl*) in *Drosophila melanogaster* but is auto-regulated in *L. cuprina*. *Sxl* also represses X chromosome dosage compensation in female *D. melanogaster*. We have developed conditional *Lctra* RNAi knockdown strains using the tet-off system. Four strains did not produce females on diet without tetracycline and could potentially be used for genetic control of *L. cuprina*. In one strain, which showed both maternal and zygotic tTA expression, most XX transformed males died at the pupal stage. RNAseq and qRT-PCR analyses of mid-stage pupae showed increased expression of X-linked genes in XX individuals. These results suggest that *Lctra* promotes somatic sexual differentiation and inhibits X chromosome dosage compensation in female *L. cuprina*. However, XX flies homozygous for a loss-of-function *Lctra* knockin mutation were fully transformed and showed high pupal eclosion. Two of five X-linked genes examined showed a significant increase in mRNA levels in XX males. The stronger phenotype in the RNAi knockdown strain could indicate that maternal *Lctra* expression may be essential for initiation of dosage compensation suppression in female embryos.

## Author summary

In the fruit fly *Drosophila melanogaster* and in the mosquito *Anopheles gambiae*, a single gene (*Sxl* in *D. melanogaster*, *fle* in *A. gambiae*) controls the development of female-specific tissues and X chromosome dosage compensation, which is the equalization of X-linked gene products in males and females. In this study we find evidence that the *transformer* gene is essential for somatic sex differentiation and repression of X chromosome dosage compensation in female sheep blow fly, *Lucilia cuprina*. In several of the transgenic

files. The nucleotide sequences of the plasmids made in this study have been deposited in GenBank. The accession numbers are MZ783066 for LctraIR and MZ783067 for LctraKI, the knockin construct. The RNA-seq reads can be accessed under Bioproject PRJNA769072.

**Funding:** This research was supported by cooperative agreements with USDA-APHIS to MJS (award numbers AP19IS000000C001and AP20IS000000C002) and by the NIFA AFRI Education and Workforce Development Predoctoral Fellowship awarded to MEW (award number 2019-67011-29550). The USDA-APHIS cooperative agreements funded part of the summer salary for MJS. The NIFA award fully supported MEW for two years. The funders had no role in study design, data collection and analysis, decision to publish, or preparation of the manuscript.

**Competing interests:** The authors have declared that no competing interests exist

strains developed, females are transformed into males on diet that lacks tetracycline. Consequently, these strains could be part of a genetic control program of this major pest of sheep in Australia.

## Introduction

X and Y sex chromosomes are thought to have evolved from a pair of autosomes [1]. As the Y chromosome degenerates over time due to a lack of recombination, X chromosome dosage compensation mechanisms have evolved to equalize X chromosome gene expression in the sexes [2]. In fruit fly *Drosophila melanogaster*, *Sex lethal* (*Sxl*) is the master regulator that controls both sex determination and X chromosome dosage compensation [3]. The female-specific SXL protein is an RNA binding protein that binds to *transformer* (*tra*) precursor RNAs and blocks the use a splice acceptor site that is used in males [4]. Consequently, only the female *tra* RNA encodes a functional TRA protein. SXL protein also binds to the 5' and 3' untranslated regions (UTRs) of the male specific lethal 2 mRNA and inhibits RNA translation [5]. MSL2 is an essential component of the MSL complex that is required for X chromosome dosage compensation [6]. Without MSL2, the MSL complex cannot assemble in females.

In the Australian sheep blow fly *Lucilia cuprina*, house fly *Musca domestica* and Mediterranean fruit fly *Ceratitis capitata*, *tra* is the central gene that controls sex determination [7–9]. Only females produce a *tra* transcript that encodes a functional protein but, unlike Drosophila, *tra* RNA splicing is autoregulated. Sex-specific splicing is initiated early in embryogenesis through maternally deposited *tra*. In male embryos, expression of a Y-linked male determining factor somehow inhibits the female-mode of *tra* RNA splicing [10,11]. The regulatory pathway downstream of *tra* is the same as in Drosophila with TRA combining with TRA2 to regulate the sex-specific splicing of *doublesex* (*dsx*) and *fruitless* (*fru*) transcripts [12–15]. DSX and FRU are DNA binding transcription factors that together control the expression of many genes that are required for sexual development and sex-specific behavior.

In *L. cuprina*, orthologs of the Drosophila X-linked genes largely remain linked but are autosomal [16]. Most of the genes that are on the X chromosome in *L. cuprina* map to the tiny fourth or dot chromosome in *D. melanogaster* [17]. X-linked genes are dosage compensated in *L. cuprina* [17,18]. The *L. cuprina no blokes* (*nbl*) gene is essential for male viability and normal X chromosome gene expression [17]. *nbl* is an ortholog of *D. melanogaster painting of fourth* (*Pof*). In Drosophila, POF binds to fourth chromosome genes and moderately enhances transcription [19]. *L. cuprina nbl* homozygous males are viable if their mothers were *nbl* heterozygotes but die at the pupal stage if mothers were homozygous for *nbl* [17]. Thus, it appears that maternal deposition of *nbl* and *tra* RNA is essential for X chromosome dosage compensation in male embryos and for sexual development in female embryos, respectively.

*Lucilia cuprina* is an ectoparasite of sheep that has a significant negative impact on sheep health in Australia and New Zealand [20]. Adult *L. cuprina* females lay their eggs on a sheep host and, after hatching, the larvae feed on dermal tissues and blood [21,22]. If untreated, the infestation can progress to severe disease and death in the most advanced cases. In Australia, the annual economic losses due reduced production and treatment costs are estimated at more than $320 million [23]. The pest is largely managed through the application of broad-spectrum insecticides. Genetic approaches such as the sterile insect technique (SIT) provide an alternative species-specific environmentally-friendly method for pest control [24]. In an SIT program, millions of insects are reared in a factory, sterilized by radiation and released at regular intervals over the targeted area. As SIT is more efficient if only males are released [25], we have

made several transgenic strains of *L. cuprina* that carry conditional female lethal genetic systems [26–28]. The embryo lethal strains carry two components; a tetracycline transactivator (tTA) driver and a tTA-activated proapoptotic effector [29,30]. Expression of tTA is driven by a promoter from a cellularization gene that has no maternal activity but is very active in early embryos. If there is no tetracycline in the maternal and larval diets, tTA will bind to the enhancer-promoter of the effector gene that contains multiple copies of the recognition site (tetO). The binding of tTA activates transcription of the *Lshid* gene, which encodes the proapoptotic *L. sericata* HID protein. However, as the *Lshid* gene contains the first intron from the *Cochliomyia hominivorax tra* gene, only the female *Lshid* RNA encodes the LsHID protein due to sex-specific RNA splicing of the *tra* intron. Rather than killing females an alternative approach would be to transform females into fertile males. That is, produce both XY and transformed XX males. Modeling suggests that, if XX males are fertile and adequately sexually competitive, this could be more efficient for population suppression than female killing [31]. Further, this approach would produce more males from each female in the mass rearing facility as all of the eggs laid will develop into males. Sex transformation could be achieved by manipulating the expression of genes required for female development such as *tra*. Disruption of the *L. cuprina tra* (*Lctra*) autoregulatory loop in female embryos by injection of *Lctra* dsRNA into precellular embryos led to partial to full sexual transformation of *L. cuprina* females [7]. Thus, we reasoned that a transgenic strain that expressed *Lctra* dsRNA could be used to produce XY and XX males for a genetic control program. Several two component strains were developed with a tTA driver and *Lctra* RNAi effector. If tetracycline was omitted from the maternal and larval diets, four double homozygous strains did not produce females. In three strains XX flies developed as intersexes or males. The fourth strain showed a high level of female transformation in double heterozygotes but double homozygous XX animals died at the pupal stage. We show that the expression of X chromosome genes is significantly elevated in these pupae. This suggests that *Lctra* may repress X chromosome hyperactivation in females.

## Results

### Knockdown of *tra* leads to masculinization of XX flies

Transgenic LctraIR effector lines were made that carry an inverted repeat of the *Lctra* transcript downstream of the $tetO_{21}$-*Lchsp70* enhancer-promoter (Fig 1A and S1 Table). The lines were initially evaluated through crossing to established "driver" or DR lines that express tTA in embryos (Figs 1B and S1) [28–30]. In the absence of tetracycline, binding of tTA to tetO activates transcription of the inverted repeat that would produce a dsRNA and trigger the native RNAi machinery within the fly [32,33], knocking down expression of *Lctra*. Of the driver lines used, only the DR3 lines show significant maternal and embryo expression [28,30]. The DR2, DR6 and DR7 lines all strongly express tTA in early embryos as each uses a promoter from a zygotic cellularization gene to drive tTA expression [27,29]. Among the offspring of these crosses, a wide range of phenotypes were observed, from no obvious transformation of females to what appeared to be full transformation (Fig 1C and 1D). The mildest phenotype was otherwise normal females with bent ovipositors. The X-linked LctraIR12X line produced offspring with the least masculinization, which was expected as the large X chromosome is mostly heterochromatic and X-linked transgenes typically show lower levels of expression [18,26] (Fig 1D). No transformation was observed among the offspring of crosses with the DR2 line (Fig 1C). For the DR6 and DR7 lines, the level of transformation observed generally correlated with the level of tTA expression in embryos [27] (Fig 1C).

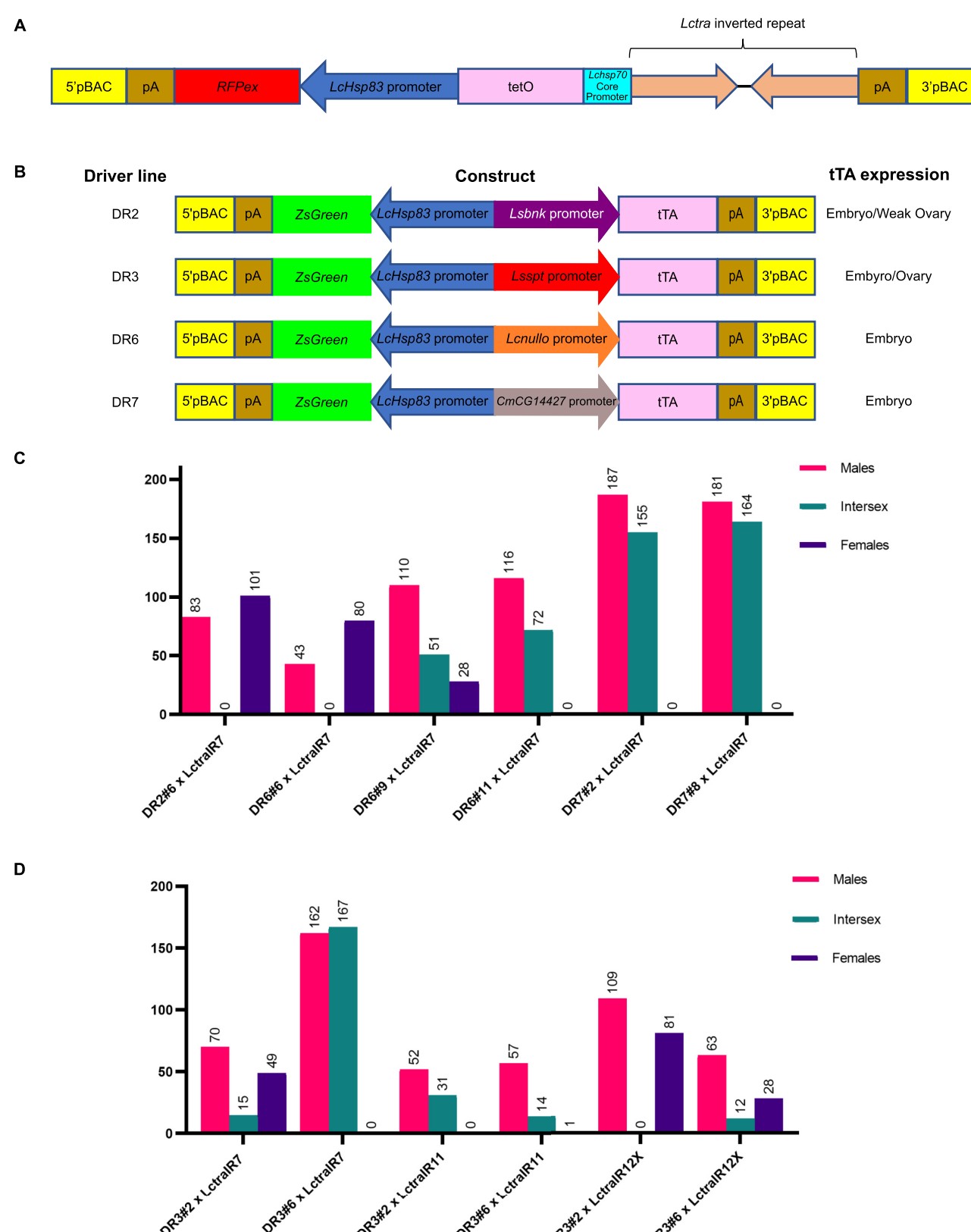

**Fig 1. Knockdown of *Lctra* leads to range of masculinization in double heterozygous XX flies. (A)** Schematic illustration of the traIR effector construct containing an inverted repeat of *Lctra* and constitutively expressed *RFPex* (DsRed-express2) marker gene. (B) Schematic illustration of the tTA driver constructs with summary of tTA stage/tissue expression. (C, D) Female to male transformation of double heterozygous flies with DR2, DR6, and DR7 (C) and DR3 (D) driver lines. Each driver and traIR line was independently derived and numbered based on the $G_0$ cross that produced transgenic offspring. Intersex flies were identified by masculinization of the female genitalia. Some intersexes also showed a decreased interocular width, which is smaller in males. One replicate is shown per cross.

Six double homozygous (DH) combinations were fertile on tetracycline and produced viable offspring on diet without tetracycline (Table 1). On diet without tetracycline, two DH strains produced some females, three strains gave a mix of males and intersexes and one strain, DR3#6; LctraIR7, produced only males (Fig 2). In the DR3#6; LctraIR7 strain, the *Lctra* RNA level was significantly reduced to 50% of wild type in mid-stage pupae (Fig 3A). To confirm masculinization was due to switching of *Lcdsx* and *Lcfru* transcripts to the male forms in this strain, mid-stage pupae were collected and sexed by PCR with Y-specific primer pairs (S2 Fig). There was a clear shift in utilizing the female splice of *Lcdsx* to the male splice in XX pupae (Fig 3B). In XX males, *Lcfru* was also predominantly splicing to the male form, with two of the three XX pupae showing only the male splice of *Lcfru*. This indicates that at the pupal stage, nearly full transformation of XX flies was likely.

## Knockdown of *Lctra* leads to hyperactivation of X chromosome expression in XX flies

The pupal eclosion rates for the DR3#6; LctraIR7 strain were on average 87% on diet with tetracycline but only 57% on diet without tetracycline (Table 1). The lower eclosion rates could indicate death of both sexes or mostly XX pupae. It appears to be the latter, as qPCR analysis of X-linked and autosomal genes of DNA from 40 adults indicated that only two were XX flies (Fig 4). The death of XX pupae could be an off-target effect of the RNAi knockdown. Alternatively, knockdown of *Lctra* mRNA could lead to hyperactivation of the X chromosomes and death of XX transformed males, as seen in *An. gambaie* females that express *femaleless* dsRNA [34]. To address these possibilities, RNAseq was conducted on mid-stage pupae that were sexed by PCR using Y-linked primer sets (S2 Fig). Of the 59 X-linked genes previously identified [17], 54 were expressed above the threshold set for this analysis (S3 Table). There was a significant increase in X linked gene expression in DR3#6; LctraIR7 XX pupae compared to XY pupae ($P = 0.002$, Mann-Whitney U Test). In comparison, there was no difference in the expression of X linked genes when comparing wildtype XX and XY pupae ($P = 0.535$, Mann-Whitney U Test) (Fig 5A). A significant difference in expression was observed in autosomal genes comparing the same sex across strains, with higher overall expression observed in XY wild type ($P < 2.2e-16$, Mann Whitney U Test) and slightly lower in XX wild type ($P = 0.002$, Mann Whitney U Test) (Fig 5B). It is unclear why autosomal RNA levels are different between the strains but as a consequence all further statistical analyses were done on the normalized RNA levels within each strain or the XX/XY ratios between strains. Consequently, we next calculated the X:A ratio, the ratio of overall expression of the X chromosome to autosomes [35,36], for each sex for both strains. The X:A ratio was significantly higher for DR3#6; LctraIR7 XX than XY pupae ($P < 0.0001$; Tukey's HSD), consistent with hyperactivation of the X chromosome (Fig 5C). The X:A ratio was higher in wildtype XX than wildtype XY (Fig 5C) as the RNA levels for the 54 X-linked genes are on average slightly higher in females than males (average female: male ratio of 1.12). We next verified these results by qRT-PCR analysis of 5 selected X-linked genes, previously shown to be dosage compensated [17,18]. The normalized RNA levels for four of the five genes were significantly higher in XX DR3#6; LctraIR7 pupae compared to XY pupae (S3A Fig). In the wild type controls, three of the five genes were

**Table 1. Rearing efficiency and female transformation of double homozygous strains.**

| Strain | Tet[a] | Rep | Pupae | Male | Intersex | Female | %AER[b] | %Male | % Intersex |
|---|---|---|---|---|---|---|---|---|---|
| DR3#6; LctraIR7 | - | 1 | 331 | 221 | 0 | 0 | 66.8 | 100 | 0 |
| | - | 2 | 166 | 78 | 0 | 0 | 46.9 | 100 | 0 |
| | - | 3 | 231 | 119 | 0 | 0 | 51.5 | 100 | 0 |
| | - | Totals | 728 | 418 | 0 | 0 | 57.4 | 100 | 0 |
| | + | 1 | 356 | 141 | 0 | 160 | 84.6 | 46.8 | 0 |
| | + | 2 | 193 | 68 | 0 | 87 | 80.3 | 43.9 | 0 |
| | + | 3 | 492 | 204 | 0 | 242 | 90.7 | 45.7 | 0 |
| | + | Totals | 1041 | 413 | 0 | 489 | 86.6 | 45.8 | 0 |
| DR3#6; LctraIR12X | - | 1 | 180 | 141 | 4 | 0 | 80.6 | 97.2 | 2.8 |
| | - | 2 | 387 | 334 | 24 | 0 | 92.5 | 93.3 | 6.7 |
| | - | 3 | 398 | 300 | 45 | 0 | 86.7 | 87.0 | 13.0 |
| | - | Totals | 965 | 775 | 73 | 0 | 87.9 | 91.4 | 8.6 |
| | + | 1 | 539 | 213 | 0 | 226 | 81.4 | 48.5 | 0 |
| | + | 2 | 406 | 157 | 0 | 167 | 79.8 | 48.5 | 0 |
| | + | 3 | 522 | 254 | 0 | 221 | 91.0 | 53.5 | 0 |
| | + | Totals | 1467 | 624 | 0 | 614 | 84.4 | 50.4 | 0 |
| DR6#9; LctraIR7 | - | 1 | 597 | 264 | 226 | 0 | 82.1 | 53.9 | 46.1 |
| | - | 2 | 492 | 213 | 158 | 0 | 75.4 | 57.4 | 42.6 |
| | - | 3 | 366 | 171 | 134 | 0 | 83.3 | 56.1 | 43.9 |
| | - | Totals | 1455 | 648 | 518 | 0 | 80.1 | 55.6 | 44.4 |
| | + | 1 | 158 | 61 | 0 | 55 | 73.4 | 52.6 | 0 |
| | + | 2 | 373 | 149 | 0 | 159 | 82.6 | 48.4 | 0 |
| | + | 3 | 114 | 47 | 0 | 43 | 79.0 | 52.2 | 0 |
| | + | Totals | 645 | 257 | 0 | 257 | 79.7 | 50.0 | 0 |
| DR3#2; LctraIR7 | - | 1 | 287 | 239 | 18 | 0 | 89.5 | 93.0 | 7.0 |
| | - | 2 | 150 | 111 | 5 | 0 | 77.3 | 95.7 | 4.3 |
| | - | 3 | 317 | 200 | 13 | 0 | 67.2 | 93.9 | 6.1 |
| | - | Totals | 754 | 550 | 36 | 0 | 77.7 | 93.9 | 6.1 |
| | + | 1 | 354 | 174 | 0 | 143 | 89.5 | 54.9 | 0 |
| | + | 2 | 541 | 251 | 0 | 240 | 90.8 | 51.1 | 0 |
| | + | 3 | 456 | 205 | 0 | 199 | 88.6 | 50.7 | 0 |
| DR6#6; LctraIR7 | - | 1 | 287 | 120 | 96 | 20 | 82.2 | 50.8 | 40.7 |
| DR3#2; LctraIR12X | - | 1 | 297 | 179 | 71 | 33 | 95.3 | 63.3 | 25.1 |

[a] "-" indicates no tetracycline in the diet and "+" indicates 100 μg/ml tetracycline in the diet

[b] %AER is the % of pupae that have eclosed into adult flies

equally expressed in males and females ($P > 0.05$), while two of the genes, *Lcmav* ($P = 0.025$) and *LcCG1909* ($P = 0.039$), were expressed higher in XY than XX pupae (S3A Fig). The ratio of female to male expression was significantly higher ($P < 0.05$) for all five genes in the DR3#6 LctraIR7 strain compared to the wild type strain (Fig 5D).

## Loss of function of *Lctra* leads to full masculinization of XX flies and increased expression of some X-linked genes

Since a loss-of-function mutation can provide independent verification of a phenotype observed from RNAi-mediated knockdown of gene activity [37], a Cas9-mediated knockin (KI) mutation was made with the third exon of *Lctra* disrupted by a *ZsGreen* fluorescent

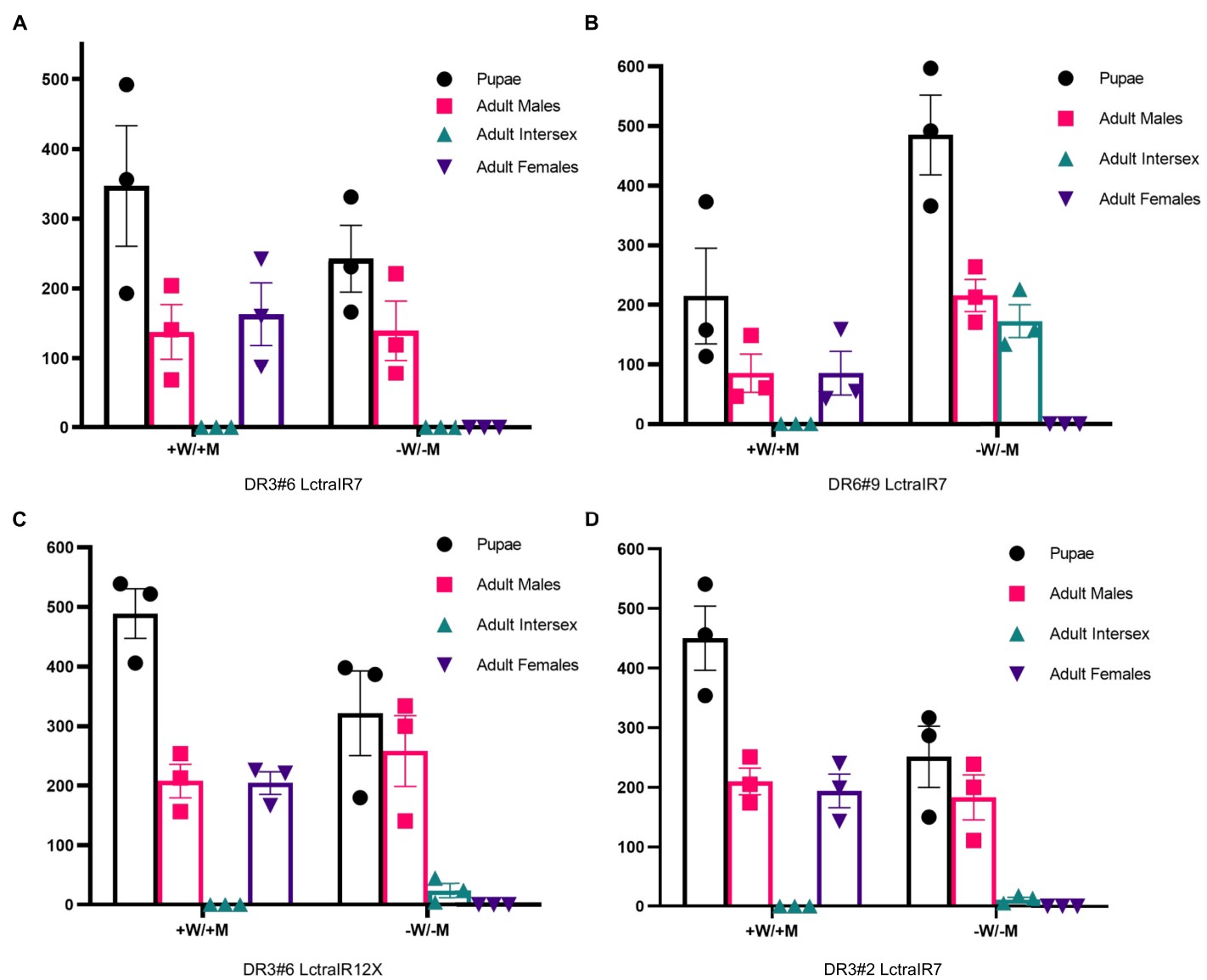

**Fig 2. Conditional knockdown of *Lctra* leads to masculinization of double homozygous XX flies.** (A-D) Female to male transformation of double homozygous lines DR3#6; LctraIR7 (A), DR6#9; LctraIR7 (B), DR3#6; LctraIR12X (C) and DR3#2; LctraIR7 (D) with (+W/+M) or without (-W/-M) tetracycline added to the maternal and larval diets. The flies were scored as male, female or intersex based on external morphology. Three biological replicates are shown with mean and standard error.

protein gene (Fig 6A). From one of the two $G_0$ males that produced fluorescent offspring (S1 Table), a KI line was established and confirmed to carry the transgene at the correct location by molecular analysis of genomic DNA (S4 Fig). The proportion of males and females were as expected based on Mendelian inheritance frequencies ($P = 0.521$, Chi-square test) (Fig 6B). Further, eclosion rates were high for each of the genotypes with a LctraKI homozygous XX male eclosion rate of 96.7%. This indicates that there was no abnormal pupal death occurring in the LctraKI XX males nor did any larval death appear to be occurring. The XX males appear to be phenotypically male in their external morphology including genitalia and interocular width (Fig 6C). Upon dissection, XX LctraKI flies had fully developed testes and accessory glands with no obvious differences from the wildtype male testes. Out of 40 LctraKI XX males single crossed with wildtype females, only 4 were fertile (10%). This suggests that the Y

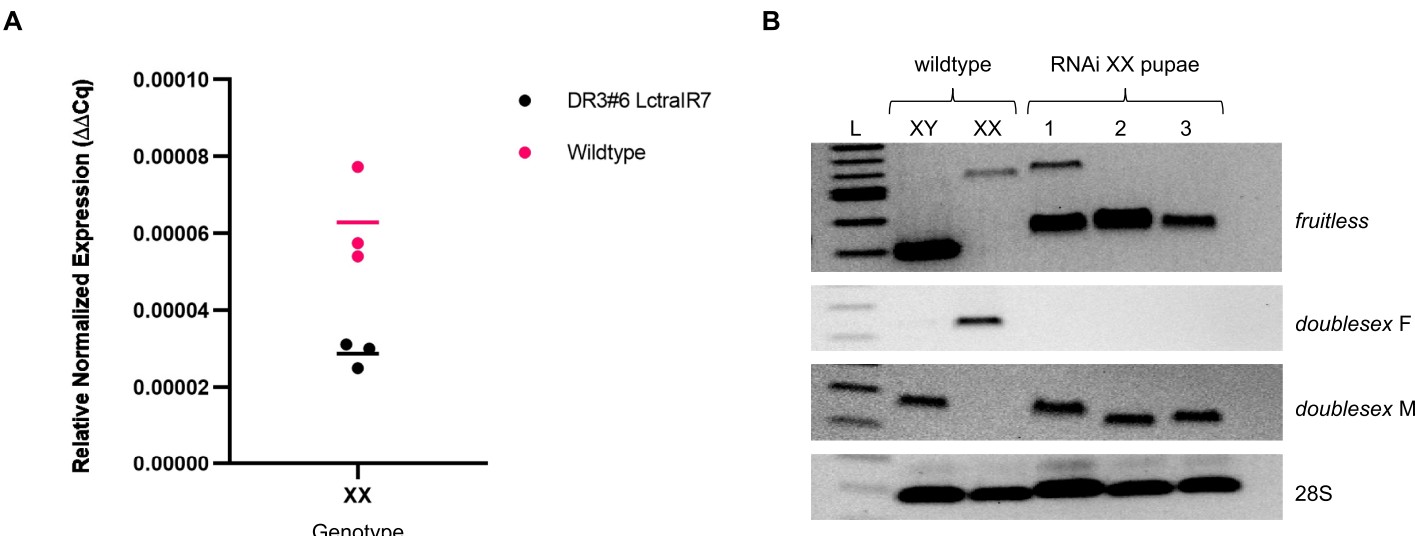

**Fig 3. Knockdown of *Lctra* in XX pupae shifts the splicing of *fru* and *dsx* primarily to the male form.** (A) qRT-PCR analysis of the RNA levels of *Lctra* in both wildtype and DR3#6; LctraIR7 XX mid-staged pupae. Expression of *Lctra* in DR3#6; LctraIR7 XX flies is significantly lower than in wildtype (P = 0.025, Student's t-test). Three biological replicates and mean expression are shown for each sample type. (B) RT-PCR analysis of *L. cuprina fruitless* and *doublesex* splicing patterns in RNAi knockdown XX mid-staged pupae compared to wildtype XX and XY controls. 28S rRNA primer set used as positive control.

chromosome may contain genes that contribute to fertility but are not essential. We next examined RNA splicing patterns in XX adults. The LctraKI had a complete switch from the female to male splice of *Lctra*, *Lcdsx*, and *Lcfru* indicating a complete transformation from female to male (Fig 6D). As XX LctraKI flies were fully viable it would appear unlikely that there was the same level of hyperactivation of X chromosome transcription as seen in XX DR3#6; LctraIR7 flies. The same five X-linked genes analyzed in the DR3#6; LctraIR7 line were analyzed by qRT-PCR. As done previously in studies on dosage compensation in adults [18,38,39], RNA was isolated from hemisected (head + thorax) LctraKI adults as most of the

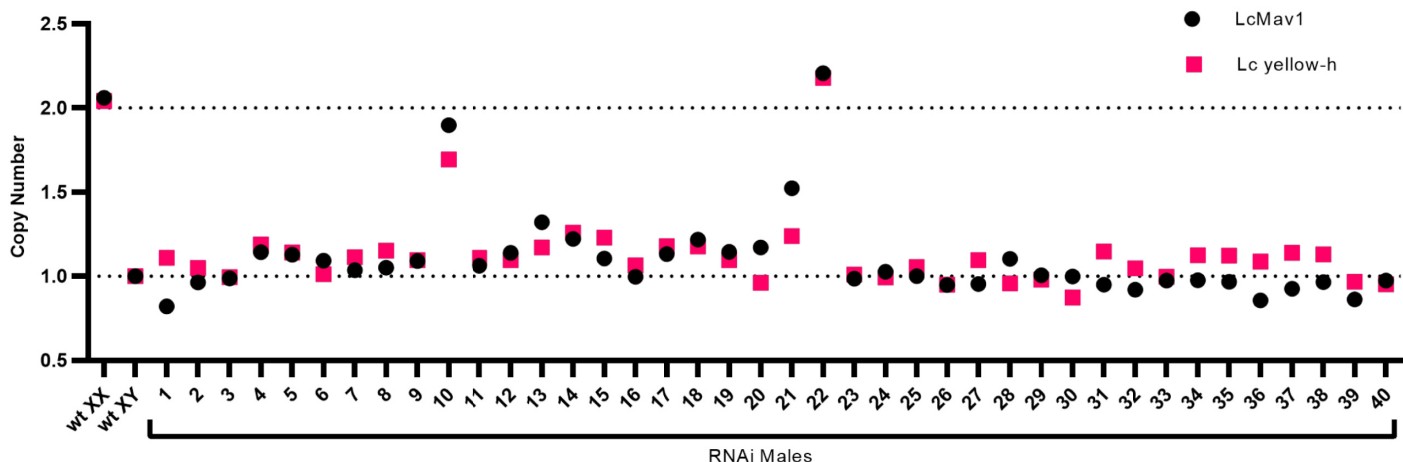

**Fig 4. Few transformed DR3#6 LctraIR7 XX flies survive to the adult stage.** qPCR analysis genotyping 40 adult DR3#6; LctraIR7 phenotypic males using two different X-linked genes- *LcMav1* and *Lc yellow-h*. RNAi males with a copy number ~1 are classified as XY while RNAi males with a copy number ~2 are classified as XX. Wildtype XX and XY adults used as control. Sample 10 and Sample 22 were found to be significantly different than the male control sample (P < 0.0001, Tukey's HSD) at both genes, while no significant difference was noted between these samples and the female controls (P > 0.05) in both genes.

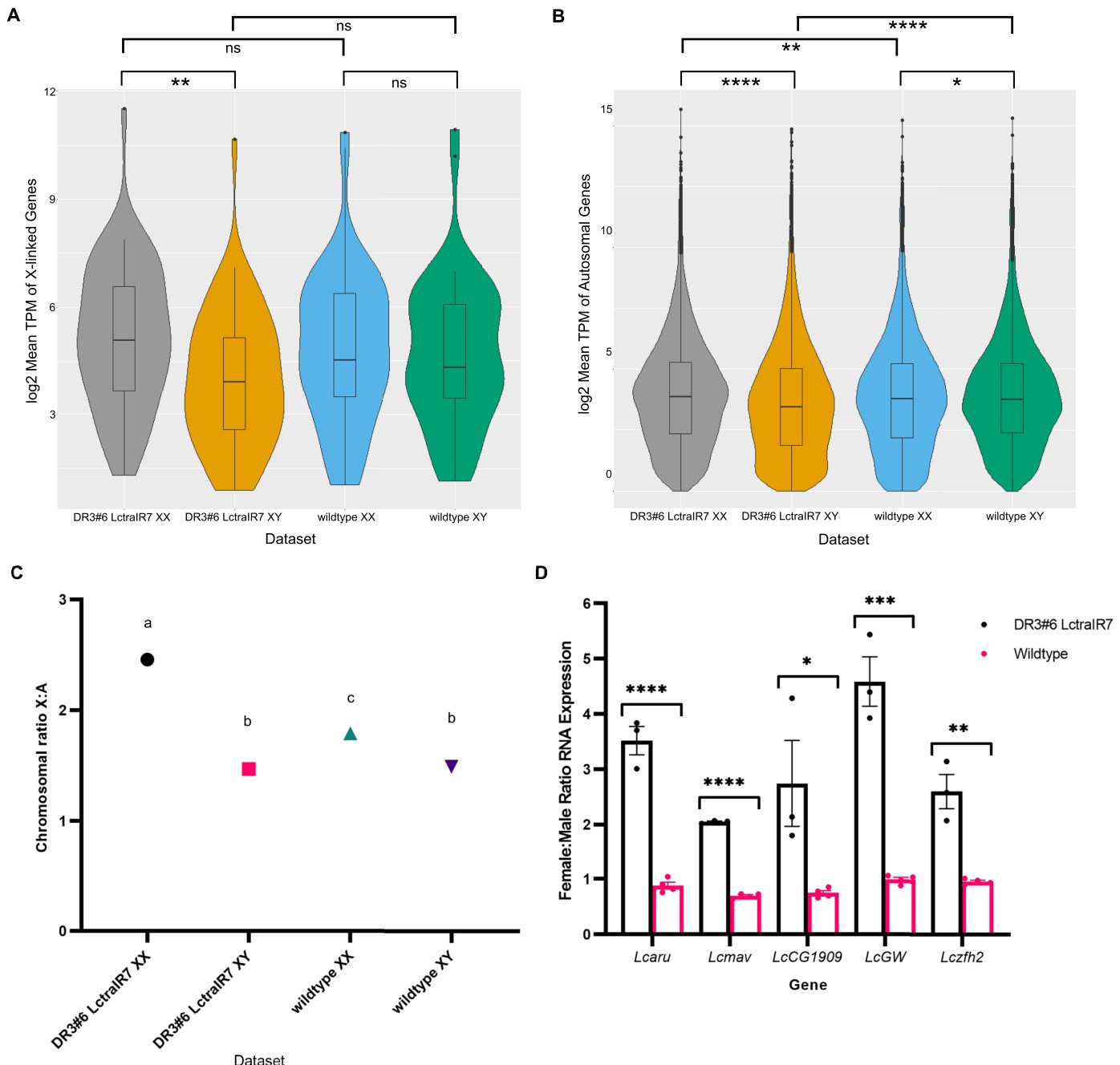

**Fig 5. Knockdown of *Lctra* leads to increased X chromosome gene expression in XX flies.** (A,B) Violin plots represent the normalized $\log_2$(TPM values + 1) of 54 X linked genes (A) or autosomal genes (B) that were above the threshold for significant expression for RNA mappings of each sex of DR3#6; LctraIR7 and wildtype to the reference. Boxplots within the violin plots represent the median and quartile ranges while the body of the violin plot shows the density of gene distribution. RNA was isolated from mid-stage pupae. The maternal and larval diets did not contain tetracycline. Significance is donated where * $P < .05$, ** $P < 0.01$, *** $P < 0.001$, and **** $P < 0.0001$ (Mann-Whitney U test). (C) Median X:A ratios for the average TPM values of each sample type. Median X:A ratios with different letters are significantly different (Tukey's HSD, $P < 0.05$). (D) qRT-PCR analysis of RNA levels for five X-linked genes shown by the ratio of XX:XY RNA expression for each strain. Biological replicates are shown as circles and bar indicates mean ratios of transcript levels in XX pupae compared to XY pupae with standard error. Significance is donated where * $P < .05$, ** $P < 0.01$, *** $P < 0.001$, and **** $P < 0.0001$ (Student's t-test).

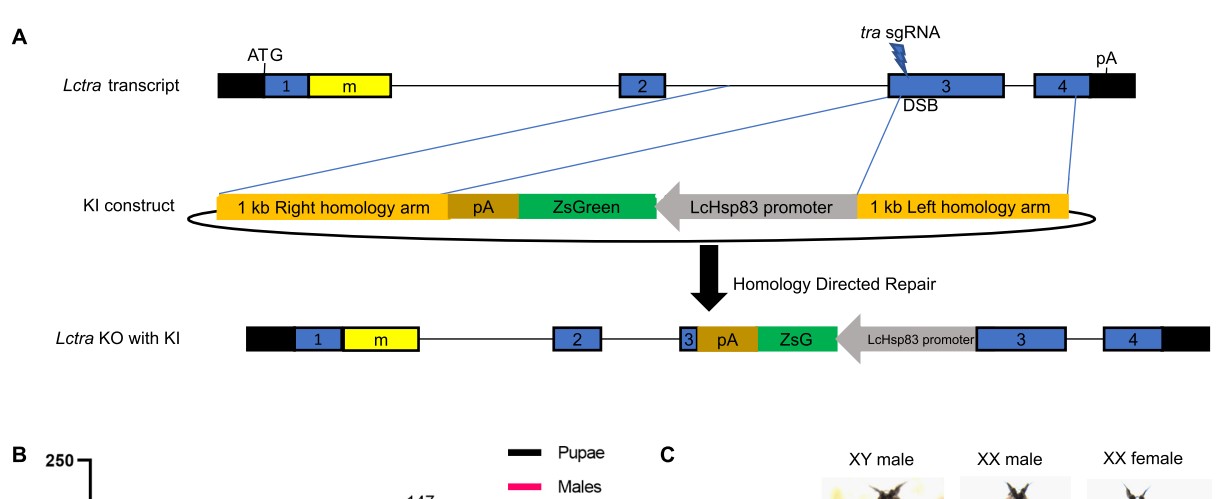

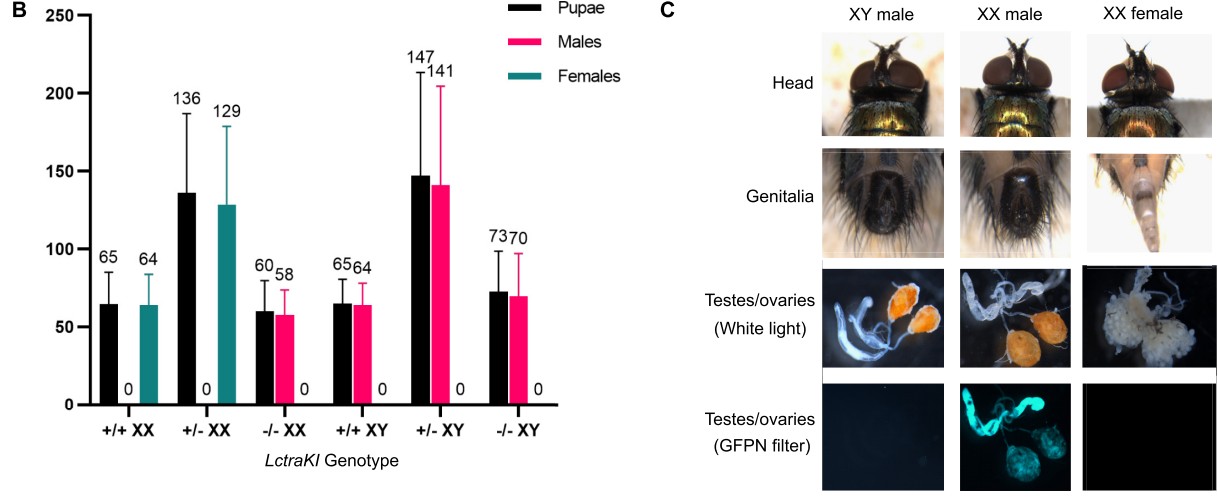

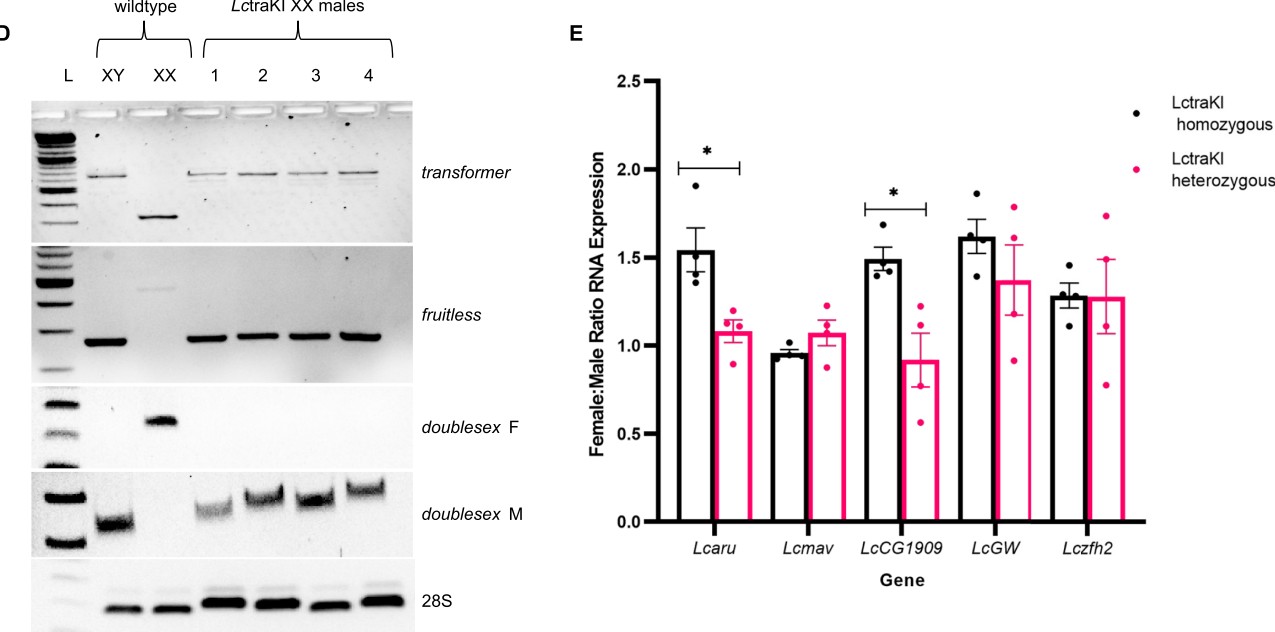

**Fig 6. Loss of function of *Lctra* leads to full masculinization of XX flies and increased expression of some X-linked genes.** (A) Schematic of the *Lctra* gene showing *Lctra* sgRNA cut site and HDR repair construct. (B) *Lctra* knockin (KI) genotype and phenotype of the offspring of a cross of males carrying an X-linked DsRed marker gene and hemizygous for the *Lctra* KI with *Lctra* KI hemizygous females. Only XX offspring inherited the red marker gene. All homozygous KI XX flies were fully transformed to males. (C) The top panels show the external morphology of wildtype XY male, *Lctra* KI XX male, and wildtype XX female including head and external genitalia. The lower panels show dissected reproductive tissue observed under white or blue light. The testes from the transgenic KI line fluoresce green. (D) RT-PCR analysis of *tra*, *fru*, and *dsx* splicing patterns in *Lctra* knockin (KI) XX males compared to wildtype XY and wildtype XX adult controls. (E) qRT-PCR analysis of RNA levels for five X-linked genes shown by the ratio of XX:XY RNA expression for LctraKI heterozygous or LctraKI homozygous hemi-sected adults. Biological replicates are shown as circles and bar indicates mean ratios of transcript levels in XX flies compared to XY flies with standard error. Significance is donated where * P < .05, ** P < 0.01, *** P < 0.001, and **** P < 0.0001 (Student's t-test).

sexually dimorphic tissue is in abdomens. The normalized RNA levels for two of the five genes were significantly higher in LctraKI homozygous XX compared to XY hemisected flies (*LcCG1909*, P = 0.015; *LcGW*, P = 0.020; Student's t-test) (S3B Fig). A third gene, *Lcaru*, shows an increase in RNA level in LctraKI XX homozygous flies but was not statistically significant (P = 0.074; Student's t-test). While not significant (P = 0.183, Student's t-test), *LcGW* expression was also noticeably higher in LctraKI heterozygous XX compared to XY siblings. This suggests that part of the increased expression seen in XX homozygotes is due to elevated expression in females at this stage of development. The female to male expression ratio was significantly elevated for two genes (*Lcaru*, P = 0.017; *LcCG1909*, P = 0.014) in the LctraKI homozygous flies compared to the LctraKI heterozygous flies (Fig 6E).

## Discussion

Maternal deposition of *tra* mRNA is thought to play an important role in the initiation of the autoregulatory loop of *tra* RNA splicing in female embryos in *L. cuprina*, *C. capitata* and *M. domestica* [7–9]. Thus, it was perhaps not surprising that *L. cuprina* DR3 driver lines that express tTA in ovaries and early embryos were particularly effective when combined with *tetO-Lctra* RNAi lines. That is, it was the knockdown of both maternal and zygotic *tra* mRNA that was effective in disrupting female development. The DR3#6/LctraIR7 combination seemed particularly potent as all XX double heterozygotes were intersexual. Further, only phenotypic males were obtained from a double homozygous DR3#6; LctraIR7 strain if tetracycline was omitted from the maternal and larval diets. Molecular analyses show that this was mostly because transformed XX males died at the pupal stage. We suggest that the death of XX pupae was due to elevated expression of X-linked genes as shown by qRT-PCR and RNAseq analyses. In *D. melanogaster* and *Anopheles gambiae* genetic control of sex determination and X chromosome dosage compensation are linked through the *Sxl* and *fle* genes, respectively. *Sxl* homozygous females die at the embryo stage due to hyperactivation of expression of X chromosome genes [40]. Similarly, transgenic *A. gambiae* lines that express *fle* dsRNA show masculinization and, in the strongest lines, death of XX mosquitoes [34]. In wild type males, the Y-linked sex determining gene *Yob* [41] is thought to somehow inactivate *fle* and thus promote dosage compensation [34,42]. In the related mosquito *A. stephensi*, expression of the Y-linked gene *Guy1* in transgenic lines was lethal to females, which appears to be due to a significant increase in expression of X-linked genes compared to autosomal genes [43]. *Guy1* is strong candidate for the male determining factor [44] and may also somehow inactivate *fle* as knockdown of *fle* expression in *A. stephensi* embryos caused female-specific lethality [34]. Our analyses of the DR3#6; LctraIR7 strain indicates that *tra* is essential for both somatic sexual development and repression of X chromosome gene expression in *L. cuprina*.

In contrast to the lethality of transformed XX males observed in the DR3#6; LctraIR7 strain, injection of *Lctra* dsRNA into precellular embryos led to full sexual transformation of some

XX individuals that were fully viable [7]. Similarly, injection of *C. capitata* embryos with *Cctra* dsRNA produced fully transformed and viable XX males [9]. In this study we showed that homozygous XX *Lctra* knockin males were fully transformed but viable. While the increase in expression of the two X-linked genes in LctraKI XX transformed males was significant, it was less than observed in DR3#6; LctraIR7 mid-staged pupae (Fig 5D). Additionally, the *Lcmav* and *Lcaru* genes were significantly increased in expression in DR3#6; LctraIR7 XX but not in XX LctraKI. This could suggest that the increased X chromosome gene expression and lethality of XX pupae observed in the DR3#6; LctraIR7 strain was due to an off-target effect and not knockdown of *Lctra* RNA. However, there are alternative explanations for why DR3#6; LctraIR7 XX transformed males died but XX males that developed from embryos injected with *Lctra* dsRNA or XX LctraKI males did not. One difference compared to the DR3#6; LctraIR7-strain is that the XX LctraKI flies are the offspring of females with one functional copy of *Lctra*. That is, the LctraKI XX males would have developed from embryos that had significant maternal deposition of TRA. Similarly, embryos injected with *Lctra* dsRNA would have wild type levels of maternally deposited TRA. Since the DR3 driver has both maternal and embryo tTA expression [27,28], the DR3#6; LctraIR7 XX flies could have had limited maternal deposition of LcTRA. It is possible that maternal LcTRA is critical for the initiation of repression of the dosage compensation machinery in XX embryos. Maternal expression of *nbl* is essential for normal X chromosome expression in male embryos [17]. The importance of maternal *Lctra* expression could be addressed by making female germline mosaics by inducing site-specific homologous mitotic recombination. This could be done with the yeast Flippase (Flp)/FRT system to mediate site-specific recombination at FRT sites that are placed in centromere-proximal regions [45]. Alternatively, CRISPR/Cas9 could be used to induce targeted mitotic recombination in the female germline [46]. Additional markers would need to be introduced onto the *Lctra* chromosome to identify germline clones homozygous for LctraKI. This approach would however fail if *Lctra* has an essential role in the female germline. Another possible explanation is that the truncated LcTRA protein that could be made in LctraKI XX flies would contain the putative amino terminal autoregulatory domain [47]. It is possible that a function of this domain is to repress dosage compensation. The domain is not found in *D. melanogaster* TRA, which has a different mechanism of dosage compensation than in *L. cuprina* [17]. This could be addressed by making additional knockin mutations that target the beginning of the *Lctra* gene.

Four DH strains were made in this study that do not produce females when raised on diet without tetracycline. These strains provide an alternative to the previously developed female-lethal strains for population suppression of *L. cuprina* [26,28–30]. In these strains, sex-specificity is achieved as the tTA-activated proapoptotic gene *Lshid* contains the first intron from the *Chtra* gene and consequently only the female transcript encodes functional protein. However, if male embryos make some of the female form of the *Lshid* RNA, then this "leaky" expression of the killing gene, while not lethal, could negatively impact male fitness. Indeed, early in development, male *C. capitata* embryos make both male and female splice forms of the *Cctra* mRNA [48]. Later in embryogenesis males make only the male form of *Cctra* RNA. It remains to be shown if the males produced by the DH strains made in this study are competitive with wild type males. The potential of the strains to control this pest could be evaluated in the future through large cage population suppression experiments [49]. Previous modeling showed that releasing males carrying a dominant female transformation gene was more effective for population suppression than fsRIDL (female-killing) [31]. This was because both XY males and the fully transformed fertile XX male offspring would pass on the dominant female transformation gene. The most effective strain made in this study, DR3#6; LctraIR7, was only partially dominant as XX flies with one copy of the female transformation system developed as sterile

intersexes. Since the XX flies are sterile the strain would not be expected to be more effective than releasing males carrying a dominant female lethal gene. Lastly, the high pupal eclosion rates and full transformation of XX males in the KI strain suggest that *Lctra* could be a good target for a homing gene drive such as developed for suppression of *An. gambiae* [50]. More complex designs such as coupling a homing gene drive targeting *Lctra* with cleave and rescue targeting a haploinsufficient gene could be also considered [51].

## Methods

### Fly rearing and germ-line transformation

The LA07 wild type strain of *L. cuprina* was maintained as previously described [18]. This wild type strain was used for *piggyBac*-mediated transgenesis as previously described [26]. Homozygous transgenic lines were maintained on a regular diet lacking tetracycline while double homozygous transgenic lines were maintained on a diet containing 100 μg/ml of tetracycline. Larvae homozygous for both transgenes were readily identified by the intensity of fluorescence of the marker genes [52]. Some double homozygous strains could not be easily made because the transgenes were located on the same chromosome. The combination DR7#8; LctraIR7 had few viable and fertile females and could not be maintained. This driver line, DR7#8, has very high tTA expression in early embryos [27]. It would appear that tetracycline supplied in the maternal diet was insufficient to fully inhibit the binding of tTA to the LctraIR effector in embryos in this strain. While females that were double homozygous with a tTA driver and the LctraIR11 effector were fertile on diet with tetracycline, embryos of both sexes did not hatch if tetracycline was omitted from the maternal diet. The embryos appeared to develop but showed extremely bright red fluorescence, indicating strong activation of the LctraIR transgene and the linked red marker gene. It is possible that other flanking genes were highly expressed in these embryos leading to lethality. Since X-linked transgenes are not dosage compensated in *L. cuprina* [18], XY and XX larvae from DR3#6; LctraIR12X strain could be separated based on fluorescence intensity of the X-linked DsRed fluorescent protein.

To make the *Lctra* KI, sgRNAs were first evaluated targeting various regions of *Lctra*. These sgRNAs were evaluated first by an *in vitro* assay to assess the efficiency of cutting the target template followed by an *in vivo* assay to determine how well they are cutting in the early embryos [53]. For the latter, $G_0$ adults were screened for phenotypic evidence of knockout. The sgRNAs targeting exons 1 and 2 were found to not be efficient at promoting *in vivo* Cas9 cleavage. The sgRNA selected that targeted exon 3 had high efficiency of cutting *in vitro* and *in vivo* and produced $G_0$ XX partially masculinized flies. An injection cocktail was made with 1.5 μl Cas9 protein (20 μM), 2.7 μl *tra* sgRNA (374 ng/μl), 2.5 μl KCl (1M), 0.5 μl phenol Red (0.5% in Dulbecco's phosphate buffered saline, Sigma catalog number P0290). This injection cocktail was incubated for 30 minutes at 37°C then 2.8 μl LctraKI construct was added and filtered through a .45 μm centrifugal filter (MilliporeSigma UFC30HV25). Pre-cellular embryos were injected and 1st instar larvae were screened for transient ZsGreen expression. All larvae with transient ZsGreen were reared to adults. Of the 20 fertile $G_0$ males that were backcrossed with wildtype virgin females, 2 produced transgenic offspring, and 1 was selected to generate a transgenic line. The insertion site was confirmed by PCR amplification with the forward primer outside of KI construct and the reverse primer within the ZsG marker of the KI construct (using primer pairs listed in S2 Table). To determine if LctraKI XX males were surviving throughout all life stages, LctraKI flies were crossed with the SLAM5X line that carries an X-linked red fluorescent protein marker gene but shows no green fluorescence [54]. Male offspring were then crossed with females that carried one copy of the LctraKI transgene. This crossing scheme allowed larvae to be scored as XX or XY based on the presence (XX) or

absence (XY) of the red fluorescent protein and as homozygous (GG), heterozygous (Gg), or wildtype (gg) for the *Lctra* KI based on the intensity of green fluorescence.

## Plasmid construction

The LctraIR constructs were built with traditional cloning strategies. All primers used for cloning can be found in S2 Table. Two sets of primers were designed to amplify a 621 bp region spanning the 3rd exon of *Lctra*. The LctraIR region was selected as it matched a single conserved exon and corresponded to most of the dsRNA previously tested in embryos [7]. Template 1 (forward portion of inverted repeat) was PCR amplified with Xho-NcoI added to the 5' end of the forward primer and AvrII added to the 5' end of the reverse primer. Template 2 (reverse portion of inverted repeat) was PCR amplified with HindIII added to the 5' end of the forward primer and NheI at the 5' end of the reverse primer. Template 1 was digested with XhoI and AvrII and cloned into pGEM-WIZ [55] cut with the same enzymes. Template 2 was digested with HindIII and NheI and cloned into the resulting plasmid from the first digestion (pGEM-WIZ-Template1) cut with the same enzymes. LctraIR was then excised using NcoI and HindIII and cloned into pBS-FL1 [26] cut with the same enzymes. The tetO-LctraIR-SV40pA cassette was then excised with NotI and XhoI and inserted into pB [Lchsp83-RFPex] [26] cut with PspOMI and XhoI.

The *tra* knock in construct was built using the NEBuilder HiFi DNA Assembly Cloning Kit following the manufacturer's protocol with all primers designed such that the amplification products overlap to facilitate assembly. Homology arms were obtained by PCR amplification of 1042 bp upstream and 928 bp downstream of the *Lctra* sgRNA cut site. The marker gene was obtained by PCR amplification of Lchsp83-ZsGreen from the DR3 plasmid [28] and the p10pA was PCR amplified from the DR6 plasmid [27]. Fragments were purified and inserted into a pBS backbone digested with HindIII and NotI.

## Female transformation assessment

To assess female to male transformation in the double heterozygous condition, 5 virgin females from a driver line were crossed in a bottle with 5 males from the LctraIR line with no addition of tetracycline to the diet. Offspring from this cross were reared on diet lacking tetracycline and adult offspring were scored as male, female, or intersex. Flies were scored as intersexual with a wide range of external phenotypes, from a bent ovipositor (~90˚ bend) but otherwise phenotypically female to fully transformed male external genitalia but female interocular distance. To assess the female to male transformation in the double homozygous condition, testing was done from established double homozygous lines reared on a maintenance level of tetracycline (100 μg/ml). Bottles were set with 5 pairs of newly eclosed flies to assess each double homozygous line either with tetracycline (100 μg/ml) or without tetracycline added to the diet. Embryos were collected and reared to adults which were scored as male, female, or intersex. To assess the female to male transformation in the L*ctra* KI line, homozygous Lctra KI males were first crossed with virgin homozygous females from a red marked X-linked line. Male offspring from this cross, containing both 1 copy of the tra KI and 1 copy of the X-linked red marker, were then crossed with virgin heterozygous tra KI females. Offspring from this cross were scored into 6 different groups based on larval fluorescence where any larvae with the red marker were XX genotype and any larvae without the red marker were XY genotype and either homozygous, heterozygous, or wildtype for the Lctra KI based on the intensity of green fluorescence. Adults from these groups were screened as male or female and expected and actual frequency of each genotype was calculated. XX LctraKI males obtained from this cross were further used for RT-PCR and qRT-PCR analysis.

## Dual RNA/DNA isolation and RNA isolations

White prepupae were collected every 20 minutes and aged for 96 h (+/- 2 h) at 25˚C, which is mid-stage of pupal development for the LA07 *L. cuprina* strain. Wildtype and DR3#6 traIR7 (-M/-W) pupae were then flash frozen in liquid nitrogen. RNA and DNA was extracted from each mid-stage pupae using the Qiagen AllPrep DNA/RNA Mini Kit following the protocol as written with the exception of increasing Buffer RLT to 1 ml and only using 100 μl of the homogenate moving forward. The remainder of the homogenate was stored at -80˚C in case higher concentrations were needed. DNA was used to genotype each pupa as XX or XY using two pairs of Y-linked primer sets (S2 Table). RNA was used for RT-PCR, qRT-PCR, and RNA-seq. Total RNA was isolated from 4 day old hemisected traKI homozygous and heterozygous adults as previously described [18].

## RT-PCR and qRT-PCR

cDNA was synthesized from 500 ng of RNA using the Superscript III First Strand Synthesis Supermix (Invitrogen) following the manufacturer's instructions. Negative controls were included which replaced the enzyme with water. cDNA was diluted 1:4 in nuclease free water. RT-PCR reactions were assembled using NEB Q5 Mastermix (*Lctra*, *dsx*F, *dsx*M, *28S*) or NEB OneTaq HS Mastermix (*fru*) using 1 μl of diluted cDNA.

qRT-PCR was conducted as previously described [18] with the exception that cDNA was generated from 500 ng of RNA instead of 3.5 μg. This change was due to the lower amount of RNA isolated from a single pupa. The primer pairs for the *L. cuprina* 28S reference gene and X-linked genes (*Lcaru*, *Lcmav*, *LcCG1909*, *Lcgw*, *Lczfh2*) were previously described [17,18]. qRT-PCR primers for evaluating *Lctra* expression were designed upstream of the LctraIR inverted repeat region (S2 Table). The raw Ct values for qPCR and qRT-PCR data shown in Figs 3–6 are provided in S4 Table.

## RNAseq analysis

RNA from individual mid-staged DR3#6; LctraIR7 and wildtype pupae were submitted to GENEWIZ, LLC for library prep, Illumina HiSeq 2 X 150 PE sequencing and initial data analysis. Quintuplicate replicates were submitted for each of the four sample types: DR3#6; LctraIR7 XX and XY and wildtype XX and XY. Libraries were generated for mRNA sequencing with polyA selection. Raw data was evaluated for sequence quality and reads were trimmed for adaptors and quality. Trimmed reads were mapped to the *L. cuprina* genome [56] and hit counts were generated for genes/exons. Differential gene expression analysis was done using DESeq2.

Additional bioinformatics were done excluding samples (DF2, DF5, and DM2) which Cook's distances and PCA analysis indicate as outliers (S5 Fig). Transcripts with less than 10 raw reads averaged among all samples were removed from analysis. Revised differential gene expression lists were generated using DESeq2 and genes with an adjusted p-value < 0.05 were called differentially expressed genes. Transcripts per kilobase (TPM) values were calculated for each transcript and averaged across all replicates of the same sample type (S3 Table).

## X to autosome (X:A) calculations

Average TPM values for all genes were calculated for each sample type. Median TPM values of all autosomal genes (TPM≥0) were calculated for each sample type. Median TPM values of all X linked genes (TPM≥0) were calculated for each sample type. Ratio of median TPM values of X chromosome to autosomes were calculated.

## Statistical analysis

For the genotyping of 40 unknown sex DR3#6; LctraIR7 phenotypically male flies, Tukey's HSD was conducted to determine if there was a difference between each unknown sample and the male and female control, independently. Unknown samples that were significantly different from the male controls ($P < 0.05$) and not significantly different from the female controls ($P > 0.05$) in both X-linked genes were classified as XX.

Mann-Whitney U Tests were first conducted to determine if there was a significant difference in overall expression level in transgenic pupae compared to wild type pupae. Although pupae were staged as accurately as possible, direct comparisons between the transgenic line and wild type are problematic as they are from different cultures.

Mann-Whitney U Tests were conducted to determine if there was a significant difference in the median TPM values of X-linked genes (Fig 5A) or autosomal genes (Fig 5B) between XX and XY samples of each strain.

For the qRT-PCR experiments, Student's t-tests were conducted for each gene to determine if there was a significant difference in RNA expression between XX and XY mid-staged pupae or hemi-sected flies of the same strain. Statistical analysis was done on the transcript levels to specifically to compare XX and XY expression levels within the same strain (S3 Fig). Additionally, Student's t-tests were conducted on the ratio of XX:XY RNA expression comparing the strains at each gene (Figs 5D and 6E).

A chi-square test was conducted on the SLAM5X/LctraKI cross to determine whether the number of adults eclosed from each genotype match the expected Mendelian inheritance pattern for the cross. The expected percentage for each genotype was as follows; +/+ XX (12.5%), +/- XX (25%), -/- XX (12.5%), +/+ XY (12.5%), +/- XY (25%), -/- XY (12.5%).

For the X:A ratios, a one-way ANOVA was conducted to compare the effects of the median X:A ratio between all 4 samples (DR3#6 LctraIR7 XX and XY, wild type XX and XY) which showed there was a difference between the median TPM values of the different samples ($P = 3.67$ e-07). This test was followed by a Post-hoc pairwise comparison using Tukey's HSD to compare two groups at a time to determine if their median X:A ratio is significant, specifically examining the DR3#6; LctraIR7 XX to XY comparison.

## Supporting information

**S1 Fig. Tetracycline-repressible female transformation system.** When strains that are homozygous for both a tTA driver and Lctra RNAi effector are raised on diet without tetracycline, tTA will bind to tetO and induce transcription of the *Lctra* inverted repeat gene. The hairpin RNA that is produced will self-anneal to form a long double-stranded RNA (dsRNA). The presence of the dsRNA will trigger an RNAi response and subsequent degradation of the *Lctra* mRNA. Without sufficient LcTRA protein, XX individuals will develop as males. The system is repressed by the addition of tetracycline to the diet, which inhibits the binding of tTA to tetO. One of the tTA drivers, DR3, has significant tTA expression in ovaries and embryos. Consequently, for production of transformed XX males, tetracycline was omitted from both the maternal and larval diets.
(PDF)

**S2 Fig. Genotyping of mid-staged pupae.** Genotyping of DNA from single mid-staged pupae DNA/RNA preps using two Y-linked primer sets. Lanes with a pink star indicate the five XX samples for both DR3#6; LctraIR7 and wild type used for RNAseq and qRT-PCR analysis. Lanes with a blue star indicate the five XY samples for both strains used in RNAseq and qRT-PCR analysis. DR3#6 LctraIR7, sample five, is missing from this genotyping as it was

collected and processed on a different day. LcGST1 was used as a control primer set, specifically as a control for determining XX flies, as the XX flies are determined by lack of a band in the Y-linked primer set PCRs.
(PDF)

**S3 Fig. X-linked gene expression levels for each independent sample type. (A)** qRT-PCR analysis of RNA levels for five X-linked genes in DR3#6; LctraIR7 or wild type XX and XY mid-staged pupae. Expression of X- linked gene is higher in DR3#6; LctraIR7 XX than XY pupae for all genes except *Lczfh2* (*Lcaru*, $P = 0.008$; *LcCG1909*, $P = 0.007$; *LcGW*, $P = 0.005$; *LcMav*, $P = 0.007$; *LcZfh2*, $P = 0.071$; Student's t-test). Expression of X-linked genes is higher in wild type XY pupae than XX pupae for two of the five genes (Lcmav, $P = 0.025$; LcCG1909, $P = 0.039$; Student's t-test) Transcript levels were normalized to 28S rRNA. Expression levels were averaged over three biological replicates (DR3#6 LctraIR7) or 4 biological replicates (wild type) and standard error shown. Significance denoted on graph as * $P < 0.05$ or ** $P < 0.01$. **(B)** qRT-PCR analysis of RNA levels for five X-linked genes in LctraKI heterozygous or homozygous XX and XY hemi-sected adults. Expression of X-linked genes in LctraKI homozygous XX compared to XY is higher in two out of the five genes tested (*Lcaru*, $P = 0.074$; *LcCG1909*, $P = 0.015$; *LcGW*, $P = 0.029$; *LcMav*, $P = 0.701$; *LcZfh2*, $P = 0.536$; Student's t-test). No significance was observed comparing LctraKI heterozygous XX and XY adults, however LcGW expression was visibly increased in LctraKI heterozygous XX flies ($P > 0.05$; LcGW, $P = 0.183$; Student's t-test). Transcript levels were normalized to 28S rRNA. Expression levels were averaged over four biological replicates and standard error shown. Significance denoted on graph as * $P < 0.05$.
(PDF)

**S4 Fig. Molecular confirmation of LctraKI location. (A)** Schematic of LctraKI into the genome. Knock in location disrupted exon 3 splitting it into two sections. Primers were designed within *Lctra* exon 2 and LcHsp83 promoter. When KI is integrated into the proper location in the genome, a band of 2714 bp will be produced. If KI is integrated into an off-target area, no band will be produced. **(B)** PCR analysis of genomic DNA from $G_1$ adult female, $G_1$ adult male, two $G_2$ pupae, and wildtype control. The $G_2$ pupae were generated from the $G_1$ male crossed with wild type females and the LctraKI line was generated from this cross. All $G_1$ and $G_2$ samples amplified the proper band (2714 bp), while the wildtype control was negative as expected. Additional non-specific banding was seen in the $G_1$ adult female.
(PDF)

**S5 Fig. Outlier detection for RNAseq samples. (A)** Principal components analysis (PCA) of all 15 samples submitted for RNA sequencing. Both XX and XY wildtype samples grouped together as expected. The DR3#6 LctraIR7 samples were not as tightly grouped, with one DR3#6 LctraIR7 XY sample appearing out of place and two DR3#6 LctraIR7 XX samples grouping together but far from the remaining three samples. **(B)** Cook's Distance Plots of each sample submitted for RNA sequencing. Samples were grouped by sample type (wild type XX, WF1-5; wild type XY, WM1-5; DR3#6 LctraIR7 XX, DF1-5; DR3#6 LctraIR7 XY, DM1-5) and lines drawn through the center of each sample type to help detect possible outliers. DF2, DF5, and DM2, together with the information from the PCA plot are considered outliers in our dataset and were not further used.
(PDF)

**S1 Table. Transformation efficiencies for generating LctraIR and LctraKI lines.**
(DOCX)

**S2 Table. Oligonucleotide primers used in this study.**
(DOCX)

**S3 Table. RNAseq supplemental data.** TPM values of autosomal and X-linked genes and X:A calculations.
(XLSX)

**S4 Table. Raw Ct values for qPCR and qRT-PCR data shown in Figs 3–6.**
(XLSX)

## Acknowledgments

We thank Fred Gould and Omar Akbari for comments on the manuscript, Clare Anstead for helpful discussions on *Lucilia cuprina* genomics, Mary Hester, Amy Berger, Scott Harrison and Nick Pistacchio for technical assistance with fly rearing, Esther Belikoff for training on injection/fly rearing, Emily Griffith for advice on statistical analysis, and Elizabeth Scholl for guidance on the RNAseq analysis.

## Author Contributions

**Conceptualization:** Megan E. Williamson, Ying Yan, Maxwell J. Scott.

**Data curation:** Megan E. Williamson.

**Formal analysis:** Megan E. Williamson.

**Funding acquisition:** Megan E. Williamson, Maxwell J. Scott.

**Investigation:** Megan E. Williamson, Ying Yan.

**Methodology:** Megan E. Williamson, Ying Yan.

**Resources:** Maxwell J. Scott.

**Supervision:** Maxwell J. Scott.

**Writing – original draft:** Megan E. Williamson, Maxwell J. Scott.

**Writing – review & editing:** Megan E. Williamson, Ying Yan, Maxwell J. Scott.

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
