## [Decision Letter · Decision Letter 0]

20 Jul 2021

Dear Dr Scott,

Thank you very much for submitting your Research Article entitled 'Conditional knockdown of transformer in sheep blowfly suggests a role in repression of dosage compensation and potential for population suppression' to PLOS Genetics.

The manuscript was fully evaluated at the editorial level and by independent peer reviewers. The reviewers appreciated the attention to an important topic but identified some concerns that we ask you address in a revised manuscript

We therefore ask you to modify the manuscript according to the review recommendations. Your revisions should address the specific points made by each reviewer.

[LINK]

Yours sincerely,

Subba Reddy Palli, Ph.D.

Associate Editor

PLOS Genetics

John Greally

Section Editor: Epigenetics

PLOS Genetics

Reviewer's Responses to Questions

**Comments to the Authors:**

Reviewer #1: This paper describes the development of transgenic strains of Lucilia cuprina, a major pest of sheep with significant economic and animal welfare impact in Australia. By knocking down Lctra, a central gene in the sex determination pathway (and for which homologues are known in a range of other insects, following original discovery and characterisation in Drosophila), the authors show that tra is necessary for dosage compensation, which is the process by which Drosophila, mosquitoes, and some other XY/XX organisms cope with having half as many copies of the X chromosome/cell in males as in females. Lctra knockdown, or mutation by knock-in, leads to various degrees of female-to-male transformation, but in stronger cases also male-type dosage compensation and death.

As the authors note in at least one regard, little of this is surprising since it generally corresponds to results from other organisms. But expecting something and seeing it are not the same thing, and this work is competently done in an economically important pest insect. Nor should one underestimate the work involved in such studies on non-model insects.

Though I am not particularly expert in this regard, I think the most novel aspect relates to the effect on dosage compensation. This pathway separates from phenotypic differentiation surprisingly (to me) high up the pathway and it is not necessarily easy to predict which regulatory genes will or will not influence it. I think that in Medfly, similar XX pseudo-males, sex-reversed by knockdown of the Medlfy tra homologue Cctra, are viable and fertile. If that is indeed the case, the authors might make more of this difference [papers from Giuseppe Saccone].

There is also a significant applied aspect to this work, as the authors note. “Sexing strains” are desirable for various genetic pest management methods, particularly sterile-male methods. The authors have previously developed such strains, and it is not obvious that the route described here would be preferable. As the authors note, a full sex-reversal system, producing fertile XX pseudo-males, could be very useful indeed, though those males would have to be sexually competitive as well as fertile (in fact for some applications sexually competitive may be more important that fertile).

In summary, a solid piece of work, no revolutionary new findings but empirical confirmation of some expectations in a demanding system of economic importance.

Minor comments

Line 60 “the master gene” probably there is no “master gene”. Tra is important, dsx is important, the various things upstream of tra are certainly important and perhaps more commonly called “the master gene”. A “key” or “central” gene, perhaps.

Line 96 I think “fertile and adequately sexually competitive…” rather than just fertile, though strictly perhaps longevity and other issues would be relevant also. This point comes up again in Discussion line 267.

Line 137 I expected Figure 2C, cited here, to show dissections – but it does not. I’m not sure that this citation of the figure supports the statement in this sentence.

Line 158 looking at the graph, expression in DR3#6… XX seems to be up relative to wild type or transgenic XY, but expression in transgenic XY appears to be slightly lower than in wild type XY. Possibly this is not significant, or perhaps there are strain differences, but perhaps worth commenting on.

Plasmids: though the construction is detailed in M&M, I would have liked to see accession numbers for the key constructs/sequences. While it should be possible to unambiguously reconstruct the sequence from the M&M data, my experience is that this is generally not be possible. To be fair, I have not tried in this particular case, so this is a general observation rather than a specific criticism

Reviewer #2: In this manuscript Williamson and colleagues studied the effect of RNAi-mediated knockdown and CRISPR/cas9-mediated knockout of transformer in the sheep blowfly. While knockdown was achieved by tet-dependent conditional expression of inverted repeats targeting transformer, knockout was achieved by knocking in a marker gene. Female to male sex conversion was observed in both knockdown and knockout experiments. Female-specific lethality was observed only in RNAi-mediated knockdown in some cases. The authors further showed that the female-specific lethality likely resulted from increased X-chromosome gene expression, linking transformer to dosage compensation. They also demonstrated the ability to produce all male progeny through female-to-male conversion or female lethality. Overall, the manuscript is well written and the work will attract the readership PLoS Genetics who are interested in either the fundamental biology of sex determination and dosage compensation or practical SIT applications. I only have a few minor points that should be addressed.

1) The authors did a good job discussing the importance of maternally deposited transformer. In this context, it will be beneficial to describe the promoters used in DR2-DR7. I may have missed it, is it true that XX lethality was only observed in one of DR3 lines, but not any line driven by the zygotic-only promoters? If so, the authors may want to clearly state that as it is consistent with their explanation as to why KI did not lead to X activation or XX lethality. Is it possible to compare the expression level of nbl in the wildtype XX, KI XX and DR3 XX embryos?

2) Figure 4: I am not sure why Y-specific primers were not used to genotype these individuals. While ddPCR works well, quantifying gene copy numbers by regular qPCR is tricky.

3) Figure 5: why is the X:A ratio in the wildtype females closer to 2 instead of 1? This does not impact the conclusion as the authors are comparing wt with knockdown lines. However, it is good to discuss.

4) I also wonder why double homozygous are needed to observe effects in the knockdown experiments. Am I mistaken?

5) Please also discuss the 2019 publication showing that a Y-linked Guy1 gene regulates dosage compensation in Anopheles stephensi by increasing the transcription of X-linked genes (Qi et al., eLife: https://elifesciences.org/articles/43570)

6) Line 264, Change “though” to “through”.

Reviewer #3: The study is well thought out and each experiment builds up towards showing that the transformer gene plays a major role in sex determination and dosage compensation in Lucilia cuprina.

Major concerns:

This study has attempted to create XX transformed males for an Australian sheep pest Lucilia cuprina. The central idea behind creating these transformed males was to introduce the parent transgenic flies in the wild as a genetic control program. The authors were successful in creating these flies but could not produce fertile transformed males implying that if a sterile insect technique to control L. cuprina population exists, these transgenic flies would create no different economic burden if released in the wild. Although their objective was not achieved, they have discovered the role of the transformer gene in L. cuprina. Further tweaking of the transgenic flies is needed to achieve the objective the authors set out for.

Specific concerns:

1. Line 93 – How does the presence of the first intron from the Cochliomyia hominivorax kill females?

2. Line 115, Figure 1A- Schematic illustration of the traIR effector construct with the Lctra inverted repeat shows a hsp83 promoter whereas the results sections says it is under hsp70 promoter.

3. Numbering system of the transgenic offspring is confusing. For eg. What does “#6” and “7” in DR3#6 LctraIR7 stand for?

4. A supplementary figure with a schematic of the cross between the DR lines and IR construct would help understand the results and the numbering system better.

5. As the four driver lines are extensively used in the paper, a brief statement on what different promoters these individual lines carry would help distinguish the driver lines.

6. Line 124 – The authors need to mention the range of intersex phenotypes they were looking for in order to clearly understand what is considered as intersex.

7. Figure4/5/6- No significance values are marked on the figures.

8. Figure3B, 6D – PCR products for doublesex M are of quite varied lengths in both the figures. A graphical representation of the band intensities in comparison to the control 28S band would clearly represent these datasets.

9. Line 195 – why were hemi-sected adults used and not the whole body for qRT-PCR?

10. Figure 5D, 6E - The X axis gene order on both the graphs is different – not making it easy to compare the two figures.

11. Line 200 - Why were the 5 genes chosen? I am not convinced that a significant increase in 2 of the chosen 5 genes translates to an hyperactivation of the X chromosome transcription in the XX DR3#6; LctraIR7 flies.

Minor concerns:

This study is based on L. cuprina being a sheep pest but nowhere have the authors mentioned why it is regarded as a pest or what the economic consequences of having it as a pest are.

**Have all data underlying the figures and results presented in the manuscript been provided?**

Reviewer #1: Yes

Reviewer #2: Yes

Reviewer #3: Yes

PLOS authors have the option to publish the peer review history of their article (what does this mean?). If published, this will include your full peer review and any attached files.

Reviewer #1: No

Reviewer #2: No

Reviewer #3: No

---

## [Editor Report · Decision Letter 1]

24 Aug 2021

Dear Dr Scott,

We are pleased to inform you that your manuscript entitled "Conditional knockdown of transformer in sheep blow fly suggests a role in repression of dosage compensation and potential for population suppression" has been editorially accepted for publication in PLOS Genetics. Congratulations!

Yours sincerely,

Subba Reddy Palli, Ph.D.

Associate Editor

PLOS Genetics

John Greally

Section Editor: Epigenetics

PLOS Genetics

Comments from the reviewers (if applicable):

**Data Deposition**

http://datadryad.org/submit?journalID=pgenetics&manu=PGENETICS-D-21-00775R1

**Press Queries**
